Adaptive prototype few-shot image classification method based on feature pyramid

Shen Linshan 1
Feng Xiang 1
Xu Li 1 xuli@hrbeu.edu.cn
Ding Weiyue 2
1 College of Computer Science And Technology, Harbin Engineering University , Harbin, HeiLongJiang , China
2 School of Mathematics, Harbin Institute of Technology , Harbin, HeiLongJiang , China
Balas Valentina Emilia
Electronic publication date: 2024 Oct 1
Publication date: 2024
Volume: 10
Electronic Location ID: e2322
Received 2024 May 10; Accepted 2024 Aug 21
Copyright: © 2024 Shen et al.
Copyright year: 2024
Copyright holder: Shen et al.
License: This is an open access article distributed under the terms of the Creative Commons Attribution License, which permits unrestricted use, distribution, reproduction and adaptation in any medium and for any purpose provided that it is properly attributed. For attribution, the original author(s), title, publication source (PeerJ Computer Science) and either DOI or URL of the article must be cited.
License URL: https://creativecommons.org/licenses/by/4.0/

Keywords: Few-shot learning, Metric learning, Prototype network, Image classification

Funding: Key Research and Development Program of Heilongjiang Province 2022ZX01A19 Fundamental Research Funds for the Central Universities 3072022TS0604 This work was funded by the Key Research and Development Program of Heilongjiang Province under Grant No. 2022ZX01A19, and by the Fundamental Research Funds for the Central Universities under Grant No. 3072022TS0604. There was no additional external funding received for this study. The funders had no role in study design, data collection and analysis, decision to publish, or preparation of the manuscript.

==============================
Few-shot learning aims to enable machines to recognize unseen novel classes using limited samples akin to human capabilities. Metric learning is a crucial approach to addressing this challenge, with its performance primarily dependent on the effectiveness of feature extraction and prototype computation. This article introduces an Adaptive Prototype few-shot image classification method based on Feature Pyramid (APFP). APFP employs a novel feature extraction method called FResNet, which builds upon the ResNet architecture and leverages a feature pyramid structure to retain finer details. In the 5-shot scenario, traditional methods for computing average prototypes exhibit limitations due to the typically diverse and uneven distribution of samples, where simple means may inadequately reflect such diversity. To address this issue, APFP proposes an Adaptive Prototype method (AP) that dynamically computes class prototypes of the support set based on the similarity between support set samples and query samples. Experimental results demonstrate that APFP achieves 67.98% and 85.32% accuracy in the 5-way 1-shot and 5-way 5-shot scenarios on the MiniImageNet dataset, respectively, and 84.02% and 94.44% accuracy on the CUB dataset. These results indicate that the proposed APFP method addresses the few-shot learning problem.

Introduction

Supervised image classification has achieved high accuracy thanks to vast amounts of data and advancements in deep learning. However, acquiring large-scale annotated datasets remains challenging, leading to increased attention on few-shot learning. Few-shot image classification necessitates training classifiers with minimal training data to recognize new categories. This poses a significant challenge as traditional classifiers typically require more labeled examples; otherwise they might suffer from underfitting or overfitting.

To address the issue of small sample sizes, the MAML method has been proposed, which can be trained on a limited number of training samples to obtain models capable of rapidly adapting to new tasks (Finn, Abbeel & Levine, 2017). Additionally, techniques such as matching networks, relation networks, and prototype networks have shown remarkable effectiveness through metric learning strategies (Vinyals et al., 2016; Sung et al., 2018; Snell, Swersky & Zemel, 2017). Furthermore, Chen et al. (2021) trained feature extractors on base-class data and achieved outstanding performance by training specific task classifiers on labeled samples from novel categories. Li et al. (2020) introduced AFHN, while Wang et al. (2020) proposed the miner network, both addressing the few-shot problem through data augmentation methods. Similarly, Baik et al. (2021) introduced the MeTAL method, and Guo & Cheung (2020) proposed the AWGIM method, employing optimization-based methods to tackle the few-shot problem. Among these, metric learning methods stand out for their simplicity and efficacy.

In recent years, metric-based methods have emerged as one of the most prominent research directions in few-shot image classification. The Feature Reconstructive Network (FRN) surpasses prior methods in both performance and computational efficiency by reconstructing query sample features and leveraging relation networks (Wertheimer, Tang & Hariharan, 2021). DeepEMD introduces an outstanding distance metric, the Earth Mover’s Distance, and achieves excellent results (Zhang et al., 2020). DeepBDC reaches state-of-the-art performance across multiple datasets by incorporating deep Brownian covariance during the learning phase to effectively learn image features, producing remarkably significant outcomes (Xie et al., 2022). The essence of metric learning methods in few-shot learning lies in acquiring exemplary feature representations and measuring the similarity between samples.

Currently, few-shot learning has also demonstrated notable achievements in various other domains. In the field of remote sensing image classification, Cheng et al. (2021) introduced a method named SPNet for few-shot image classification, which effectively enhances the accuracy of remote sensing image classification through prototype self-calibration and intercalibration. In the realm of few-shot image segmentation tasks, Cheng, Lang & Han (2022) proposed a Holistic Prototype Activation (HPA) network as a solution to the challenges of overfitting and imprecise segmentation boundaries encountered in few-shot segmentation. Furthermore, Lang et al. (2023) proposed a new method called BAM, which uses base learners to identify basic regions in query images and point clouds that are easily confused, effectively alleviating the bias issue of the FSS model towards seen concepts in 2D and 3D scenes, thereby achieving significant results. Additionally, Lang et al. (2024) introduced the DCP method, which utilizes a divide-and-conquer strategy in few-shot segmentation tasks. This method divides coarse results into smaller regions and addresses segmentation failures by utilizing information obtained from support image-mask pairs (Lang et al., 2024).

This study draws inspiration from the Feature Pyramid Network (FPN) (Li et al., 2020) to enable the feature extractor to learn an optimal feature representation. It proposes a novel feature extractor, FResNet, integrating ResNet architecture and feature pyramid structure. FResNet aims to facilitate learning more diverse sample features. The method combines the strengths of ResNet architecture and feature pyramid structure, enabling simultaneous extraction of image features and better capturing local and global object information, thereby enhancing target classification performance. Additionally, addressing inaccuracies stemming from the traditional method’s use of a fixed class prototype to represent the support set, this article introduces an Adaptive Prototype method (AP). This approach extensively leverages information from query samples when computing class prototypes for the support set, allowing for the computation of optimal support set class prototypes based on the current query sample. This method aids in more accurately selecting representative samples for class prototypes and calculating their similarity.

This article introduces an Adaptive Prototype few-shot image classification method based on the Feature Pyramid (APFP) to facilitate optimal feature representation learning by the feature extractor. The APFP comprises the FResNet network and the Adaptive Prototype method. The FResNet method amalgamates the advantages of the ResNet architecture and the feature pyramid structure. It enables the concurrent extraction of image features and better capture of both local and global object information, thereby enhancing target classification performance. The Adaptive Prototype method extensively utilizes information from query samples when computing class prototypes for the support set. This allows for the computation of optimal class prototypes based on the current query sample. This method enables a more accurate selection of class prototypes for the support set than the standard Mean Prototype (MP) method.

This article’s contributions can be summarized as follows: The article introduces a novel feature extractor utilizing an improved feature pyramid structure, termed FResNet. Results demonstrate a significant enhancement in performance for similarity measurement in few-shot image classification when employing the designed feature extractor.

The article proposes an Adaptive Prototype method within the prototype network framework. This method adaptively selects corresponding prototype samples for similarity measurement based on different query samples. Experimental results show substantial improvements on multiple few-shot learning benchmark datasets using the proposed method.

The article conducted extensive experiments on the few-shot classification datasets MiniImageNet and CUB to demonstrate the effectiveness of the proposed APFP method. Experimental results indicate that the APFP method demonstrates outstanding performance.

Related works

Few-shot learning

In the domain of few-shot learning tasks, each category is represented by a finite number of training samples, collectively referred to as the support set. Classification of the query set is achieved by training models on this limited support set. However, employing a sparse set of samples for model training may lead to overfitting issues, resulting in suboptimal performance. Presently, few-shot learning tasks can be roughly categorized into four distinct types: data augmentation, metric learning, parameter optimization, and other methods. Among them, metric learning methods address the problem of learning reliable classification models with limited sample sizes by training models capable of effectively comparing distances between samples. The primary concept underlying metric learning methods is to classify samples by calculating the distances between samples and different classes using a specific metric. Specifically, this involves computing the Euclidean distance between query samples and support set samples or employing nonlinear methods such as neural networks to calculate the similarity between query samples and support set samples, thereby obtaining classification results.

In metric learning within few-shot learning, the primary objective is to construct a feature extractor capable of mapping input images into a feature space while defining a distance or similarity function. The design of this function aims to minimize the distance between samples with the same label in the feature space, thereby promoting their similarity while maximizing the distance between samples with different labels, facilitating better discrimination among them. The advantage of this metric learning approach lies in its ability to effectively utilize a small number of samples to establish meaningful category boundaries, thus demonstrating robust generalization capabilities when faced with new categories. For example, classical prototype networks utilize neural networks to map each sample into a shared space, where the mean of samples from each class serves as the prototype for that class. Subsequently, the Euclidean distance between query samples and each prototype is computed as a similarity score (Snell, Swersky & Zemel, 2017). Similarly, relation networks employ a neural network as the metric module to compute the similarity between query samples and support set samples (Sung et al., 2018). Such learning processes facilitate better discrimination and inference of the similarities and differences between different categories. Many recent state-of-the-art methods have also achieved outstanding results by adopting the principles of metric learning. For example, DeepEMD (Zhang et al., 2020) utilizes the Earth Mover’s Distance as a metric, while DeepBDC (Xie et al., 2022) also employs the metric-based method of prototype networks. For metric learning, the key lies in designing an excellent feature extractor to learn the optimal feature representation of input samples. Another crucial aspect is selecting a representative class prototype for metric computation.

Feature pyramid

The feature pyramid is a crucial technique for image feature extraction, capable of capturing features at multiple scales to adapt to objects and scenes of different sizes. Particularly noteworthy in this field is the FPN (Feature Pyramid Network), introduced by Lin et al. (2017), which integrates feature pyramids from different levels to obtain multi-scale feature representations within a single network. Specifically, FPN leverages top-down feedback to combine high-level semantic information with low-level detailed information, achieving rich and accurate feature representations at different scales. This effective extraction of multi-scale features has led to the widespread application of FPN in computer vision tasks such as object detection and semantic segmentation, resulting in significant performance improvements. In recent work on few-shot image classification, SetFeat utilizes feature pyramids to propose a feature set extraction and matching method, achieving excellent results (Afrasiyabi, Lalonde & Gagné, 2020). Inspired by the FPN network structure, this article proposes the FResNet algorithm for obtaining multi-scale feature representations of images and integrating them during the feature extraction stage.

Prototype network

The prototype network is a classic model in few-shot learning, which computes class prototypes from support set samples for classification. During training, the prototype network takes the support set as input and calculates prototype vectors for each class, typically using the mean of class samples as the prototype. During inference, the similarity between query samples and each class prototype is computed using Euclidean distance. As a classic method in metric learning, the prototype network has provided essential insights for many subsequent methods. However, it has limitations; sample distributions are often uneven, and using the mean of class samples as prototypes may not effectively represent class prototypes. To address this, Zhou & Yu (2023) proposed WPN, which utilizes graph neural networks to measure the contribution of same-class samples to their prototypes. Liu, Song & Qin (2020) proposed BD-CSPN, which eliminates the difference between prototype estimates and ground truth through two bias-eliminating mechanisms. Zhang & Huang (2022) proposed DSFN, which reduces bias induced by intra-class variations through spectral filtering. This article proposes a lightweight Adaptive Prototype method, which adaptively computes prototypes for each class based on information from query samples and support set samples, achieving excellent results.

Methods

In few-shot learning tasks, the N-way K-shot paradigm represents the task type, wherein the query set comprises N classes, each with K samples. This article considers two datasets: a support set S={(xi,yi)}i=0N×K and a query set Q={(xj)}j=0M. The support set S contains labeled samples representing known categories. In contrast, the query set Q comprises unlabeled samples for which the labels need to be inferred using the support set S. Here, xi and yi denote samples and their corresponding labels for the known categories, respectively, while xj represents samples with unknown labels sampled from the query set. N signifies the presence of N distinct categories in the support set S, K denotes the number of samples per category in the support set S, and M represents the number of samples in the query set Q. The objective of this article is to utilize the support set S under the N-way K-shot framework to predict the category of a given query sample xq∈Q.

This article adopts the prototype network paradigm to address the task of few-shot image classification. Initially, the support set S and query set Q of the few-shot classification task are separately fed into a feature extractor. In this regard, the selected feature extractor is FResNet—an embedding network designed in this article that combines the FPN structure to capture local and global objects’ information more effectively. Subsequently, an Adaptive Prototype algorithm is used to compute prototypes for the sample features obtained from the feature extractor. Then, the Euclidean distances from the query vectors to the prototypes of each class are calculated, and classification results are derived based on these distances. The overall network architecture of this article is illustrated in Fig. 1. Subsequent sections will provide a detailed exposition of the methodology adopted in this article.

Figure 1 The overall structure of the method.

This study’s 5-shot image classification framework consists of the following steps: employing FResNet to extract image features and obtain feature vectors for support set samples and a query sample. The Adaptive Prototype module calculates the distances, denoted as di, between support set samples and the query sample, generating class prototypes accordingly. Finally, the Euclidean distance is utilized to measure the distance between the support set class prototypes and the query sample, facilitating the classification process. Figure source credit: CUB dataset.

FResNet extractor

ResNet (He et al., 2016) is often the preferred choice as a feature extractor in metric learning tasks. DeepBDC (Xie et al., 2022) employs ResNet-12 (Tian et al., 2020) and ResNet-18 (Liu et al., 2020), while FRN (Wertheimer, Tang & Hariharan, 2021) opts for ResNet-12 as its feature extractor. ResNet and its variations are better suited for learning features on limited datasets due to their improved gradient propagation. Despite ResNet’s effectiveness in learning dataset features, it fails to capture rich, fine-grained details within images in few-shot scenarios. To address this issue, the present study enhances the original ResNet architecture by introducing FResNet as an augmentation to ResNet. Drawing inspiration from the feature pyramid structure in FPN (Lin et al., 2017), an innovative backbone network structure named FResNet is introduced. The approach presented in this article builds upon the ResNet architecture and is enhanced accordingly. Specifically, a feature pyramid is constructed at the output of the last residual block in each stage of ResNet to obtain outputs containing features at different levels. In the design of the feature pyramid, this article opts to utilize the feature vectors from the lowest level as the final output of the proposed feature extractor, FResNet. This decision is driven by the understanding that the feature vectors from the lowest level typically preserve more detailed information and exhibit superior capability in capturing both local and global features of objects in images. Ultimately, the feature vectors output by the FResNet extractor are utilized for subsequent similarity measurement and classification tasks. The specific implementation of FResNet is detailed below.

In Fig. 2, the computational path of the proposed FResNet is illustrated. This path comprises two stages: initially, the bottom-up computation of sample feature maps, followed by the top-down computation of the feature pyramid’s output. During the bottom-up feature extraction phase, ResNet is employed for forward computations. The feature pyramid is constructed using the output feature maps of the last three residual blocks {C1,C2,C3} from the ResNet. These features typically encompass more abstract and high-level information, with C3 representing the topmost level of the pyramid, carrying the most robust semantic features, followed by C2 and C1.

(1) {C1,C2,C3}=fresnet(x)

Figure 2 Computational diagram of FResNet.

It records the intermediate features obtained from different residual blocks in the ResNet network, enhances these intermediate features through an attention module, resizes the scales using the nearest method, and integrates features from different levels using the concatenation method. Figure source credit: CUB dataset.

Initially, in computing the feature pyramid output from top to bottom, each output feature map undergoes processing through an attention module, capturing correlations between different positions or channels and emphasizing critical elements while attenuating irrelevant features. Subsequently, for higher-level features, a straightforward and effective nearest-neighbor upsampling method is employed to match the scale, expanding the feature map dimensions to enhance resolution and retain original information, thereby providing a more nuanced feature representation for the pyramid. Lastly, the high and low-level features are concatenated across channels, integrating advanced semantic information with high-resolution local details to generate a more comprehensive and enriched feature representation.

(2) P1=fnearest(fSAM−3(C3))

(3) P2=concat(fSAM−2(C2),P1)

(4) P3=concat(fSAM−3(C1),fnearest(P2))

The process of constructing the feature pyramid in this article is depicted by Eqs. (2)–(4). Eventually, three distinct-scale, different-level features P1,P2,P3 are obtained. By fusing features from different levels, the feature map P3 encompasses more detailed and semantic information from the image. This study designates P3 as the output of FResNet, representing an aggregation of multi-level feature representations. Subsequently, P3 is fed into the metric module for similarity comparison and classification purposes.

Adaptive prototype

Most current methods follow the methodology of prototype networks (Snell, Swersky & Zemel, 2017), using the mean of individual samples as class prototypes. This method assumes that samples are based on Bregman divergences, where the points in the feature space closest to the mean of these samples are considered mean points. However, a singular mean is highly sensitive to outliers or noisy samples in few-shot learning. This sensitivity might lead to inaccurate class prototypes influenced by outliers, while sample distributions tend to be more diverse and uneven. Using a simple mean as a class prototype might not sufficiently capture this diversity. Therefore, this article introduces an innovative method to determine class prototypes, aiming to model them more accurately. Class prototypes are dynamically calculated by considering the correlation, diversity between samples, and uncertainty of the underlying distribution, which can represent categories more comprehensively and accurately. The method proposed in this article provides an innovative solution to the metric learning task in few-shot learning. The comparison between the adaptive prototype method and the mean prototype method is shown in Fig. 3. The Adaptive Prototype method proposed in this article is detailed as follows.

(5) Di=(si−xq)2

Figure 3 Comparison between mean prototype and adaptive prototype.

This article proposes an Adaptive Prototype method that dynamically calculates class prototypes based on the similarity between support and query samples. Different query samples generate different prototypes for the same class of support samples rather than using a single prototype for all query samples. In contrast, the Mean Prototype method only calculates the mean of the support samples and does not customize the class prototype to fit the query samples.

This article acknowledges the diverse and uneven characteristics present within support set samples in few-shot learning. It asserts that directly using the mean of samples as class prototypes for all query samples is inaccurate. The process of computing class prototypes via sample means risks losing the diversity within the class. Consequently, the classification performance is significantly compromised if the features specific to a class present in query samples are lost in the class prototype. Hence, this article introduces a novel method that adaptively computes class prototypes based on the features of the current query sample. The proposed method selects more relevant support set samples by the features of the query sample, assigning higher weights to these samples while assigning lower weights to the remaining ones, thus deriving the class prototype. This article initially computes the Euclidean distance between the query sample xq∈Q and each support set sample si∈S, representing the similarity between the query sample and each support sample. The calculation formula is depicted in Eq. (5). Then, employing the softmax function, this study computes the weights for each sample as depicted in Eq. (6).

(6) wj=exp(−Dj×θ)∑i=1kexp(−Di×θ).

In Eq. (6), wj represents the weight of the support set sample si relative to the current query sample xq. Due to the potential presence of samples in the support set that deviate from the typical class prototype, referred to as outlier samples, these samples may lead to deviations in the computed class prototype. To address this issue, a hyperparameter θ is introduced during the computation process to mitigate the influence of outlier samples on the class prototype. The parameter θ in Eq. (6) can reduce the weights of outlier samples. By adjusting the value of θ, the influence of these outliers on the class prototypes can be effectively mitigated. Multiple experiments determined that the optimal value of θ for the mini-ImageNet dataset is 0.008, while for the CUB dataset, the optimal value is 0.072. Consequently, these values are set for subsequent calculations to 0.008 and 0.072, respectively. Finally, this study utilizes the weights obtained for each sample to compute the class prototypes.

(7) Xproto=∑j=1k(sj×wj).

In Eq. (7), Xproto represents the class prototypes obtained from the support set. This article uses the Euclidean distance to measure the similarity between the query sample and each class prototype. This captures the core idea of the proposed Adaptive Prototype method in this article.

Experiments

Dataset

MiniImageNet. MiniImageNet was first introduced in Vinyals et al. (2016). It is a popular dataset used in few-shot learning research based on the ImageNet dataset. Comprising a collection of images, MiniImageNet categorizes these images into 100 different classes. To conduct few-shot learning tasks, MiniImageNet divides its image data into various subsets, including sets designated for model training, validation, and testing. This dataset has emerged as a critical resource in the field of few-shot learning, extensively employed for evaluating and comparing the performance of various learning algorithms.

CUB-200-2011. The CUB dataset is a crucial dataset employed in few-shot learning tasks, initially designed for fine-grained image classification purposes (Wah et al., 2011). Originally utilized for fine-grained bird classification, it encompasses approximately 11,788 images originating from 200 distinct bird species. These images typically exhibit high resolution and come annotated with detailed information, such as species of the birds. In the realm of few-shot learning, the CUB dataset presents challenges due to subtle differences between each category. Following the partitioning outlined in Vinyals et al. (2016), this article divides the 200 categories into 100, 50, and 50 for meta-training, meta-validation, and meta-testing purposes. The CUB dataset is pivotal in advancing research within the domain of few-shot learning.

Implementation setting

Architecture. This article adopts DeepBDC as the baseline. Employing DeepBDC’s fundamental structure, this study utilizes FResNet based on ResNet-12 for feature extraction on the MiniImageNet dataset and ResNet-18-based FResNet on the CUB dataset. Consistent with typical tasks, image input resolutions are set to 84 × 84 and 224 × 224, respectively. When computing class prototypes, the article replaces the previous method with its Adaptive Prototype method.

Training. The model in this article undergoes episodic training within a meta-learning framework. Initially, the model is pre-trained with initialized weights. Subsequently, each episode (task) involves standard 5-way 1-shot and 5-way 5-shot classification training. The embedding model network is then trained on the entire meta-training set, encompassing all categories, using cross-entropy loss.

Algorithm 1 Calculate similarity.

Input: Support set S={s1,s2,…,sn}, Query sample xq	
Output: Similarity score Score	
1: n← size of S                             ⊳ Get the size of support set S	
2: W←0                                     ⊳ Initialize total weight W	
3: θ←0.008                                     ⊳ Set hyperparameter θ	
4: Xproto←0                          ⊳ Initialize prototype vector Xproto	
5: if n>1 then              ⊳ If support set contains more than one sample	
6:   for i←1 to n do             ⊳ Loop over each sample in the support set	
7:     Di←(si−xq)2                 ⊳ Compute Euclidean distance Di	
8:     W←W+exp(θ×Di)                  ⊳ Calculate total weight W	
9:   end for	
10:  for i←1 to n do         ⊳ Loop over each sample in the support set again	
11:     wi←exp(θ×Di)/W                      ⊳ Compute weight wi	
12:    Xproto←Xproto+wi×si            ⊳ Update prototype vector Xproto	
13:  end for	
14:   Score←(Xproto−qi)2                   ⊳ Calculate similarity score	
15: else                             ⊳ If support set contains only one sample	
16:   Score←(s1×qi)          ⊳ Calculate similarity score for single sample	
17: end if	

Experimental results

This article evaluates the accuracy of the proposed FResNet and Adaptive Prototype on the MiniImageNet and CUB datasets. Meta-DeepBDC is chosen as the baseline for comparison against recent popular few-shot classification methods, most of which fall under metric-based methods. Two commonly used benchmarks, namely 5-way 5-shot and 5-way 1-shot, were selected for evaluation in this study. Experimental results on the MiniImageNet dataset are presented in Table 1, while results on the CUB dataset are shown in Table 2.

Table 1 Experimental results of the proposed APFP method on the MiniImageNet dataset are presented.

Bold indicates the best experimental results, while italic indicates the second best. § denotes results reproduced using the settings outlined in this article.

Method	Backbone	5-way 1-shot	5-way 5-shot	
PPA (Qiao et al., 2018)	WRN-28-10	59.60 ± 0.41	73.74 ± 0.19	
wDAE-GNN (Gidaris & Komodakis, 2019)	WRN-28-10	62.96 ± 0.15	78.85 ± 0.11	
LEO (Rusu et al., 2019)	WRN-28-10	61.76 ± 0.08	77.59 ± 0.12	
SetFeat (Afrasiyabi et al., 2022)	SF-12	68.32 ± 0.62	82.71 ± 0.46	
ProtoNet§ (Snell, Swersky & Zemel, 2017)	ResNet-12	61.83 ± 0.44	80.05 ± 0.31	
FEAT (Ye et al., 2020)	ResNet-12	66.78 ± 0.20	82.05 ± 0.14	
Meta-Baseline (Chen et al., 2021)	ResNet-12	63.17 ± 0.23	79.26 ± 0.17	
WPN (Zhou & Yu, 2023)	ResNet-12	–	70.37 ± 0.64	
BD-CSPN (Liu, Song & Qin, 2020)	ResNet-12	65.94	79.23	
DSFN (Zhang & Huang, 2022)	ResNet-12	61.27 ± 0.71	80.13 ± 0.17	
MELR (Fei et al., 2021)	ResNet-12	67.40 ± 0.43	83.40 ± 0.28	
FRN (Wertheimer, Tang & Hariharan, 2021)	ResNet-12	66.45 ± 0.19	82.83 ± 0.13	
IEPT (Zhang et al., 2021)	ResNet-12	67.05 ± 0.44	82.90 ± 0.30	
BML (Zhou et al., 2021)	ResNet-12	67.04 ± 0.63	83.63 ± 0.29	
DeepEMD (Zhang et al., 2020)	ResNet-12	65.91 ± 0.82	82.41 ± 0.56	
Distill (Tian et al., 2020)	ResNet-12	64.82 ± 0.60	82.14 ± 0.43	
DMF (Xu et al., 2021)	ResNet-12	67.76 ± 0.46	82.71 ± 0.31	
APP2S (Ma et al., 2022)	ResNet-12	66.25 ± 0.20	83.42 ± 0.15	
MLCN (Dang et al., 2023)	ResNet-12	65.54 ± 0.43	81.63 ± 0.31	
TAPR (Zhang & Gu, 2023)	ResNet-12	66.04 ± 0.64	82.23 ± 0.40	
Meta-DeepBDC (Xie et al., 2022)	ResNet-12	67.34 ± 0.43	84.46 ± 0.28	
APFP (ours)	ResNet-12	67.98 ± 0.44	85.32 ± 0.28	

Table 2 Experimental results of the APFP method proposed in this article on the CUB dataset are presented.

Bold indicates the best experimental results, while italic indicates the second best. § denotes results reproduced using the settings outlined in this article. * taken from Tang, Huang & Zhang (2020).

Method	Backbone	5-way 1-shot	5-way 5-shot	
ProtoNet (Snell, Swersky & Zemel, 2017)	Conv4	64.42 ± 0.48	81.82 ± 0.35	
FEAT (Ye et al., 2020)	Conv4	68.87 ± 0.22	82.90 ± 0.15	
MELR (Fei et al., 2021)	Conv4	70.26 ± 0.50	85.01 ± 0.32	
WPN (Zhou & Yu, 2023)	Conv4	–	87.03 ± 0.65	
SetFeat (Afrasiyabi et al., 2022)	SF-12	79.60 ± 0.80	90.48 ± 0.44	
MatchNet (Vinyals et al., 2016)	ResNet-12	71.87 ± 0.85	85.08 ± 0.57	
BD-CSPN (Liu, Song & Qin, 2020)	ResNet-12	84.90	90.22	
MLCN (Dang et al., 2023)	ResNet-12	77.96 ± 0.44	91.20 ± 0.24	
AA (Afrasiyabi, Lalonde & Gagné, 2020)	ResNet-18	74.22 ± 1.09	88.65 ± 0.55	
ProtoNet§ (Snell, Swersky & Zemel, 2017)	ResNet-18	80.85 ± 0.43	89.95 ± 0.23	
MAML* (Finn, Abbeel & Levine, 2017)	ResNet-18	68.42 ± 1.07	83.47 ± 0.62	
Neg-Cosine (Liu et al., 2020)	ResNet-18	72.66 ± 0.85	89.40 ± 0.43	
LaplacianShot (Ziko et al., 2020)	ResNet-18	80.96	88.68	
Baseline++ (Chen et al., 2019)	ResNet-18	67.02 ± 0.90	83.58 ± 0.54	
FRN (Wertheimer, Tang & Hariharan, 2021)	ResNet-18	82.55 ± 0.19	92.98 ± 0.10	
AAP2S (Ma et al., 2022)	ResNet-18	77.64 ± 0.19	90.43 ± 0.18	
Meta-DeepBDC (Xie et al., 2022)	ResNet-18	83.55 ± 0.40	93.82 ± 0.17	
APFP (ours)	ResNet-18	84.02 ± 0.40	94.44 ± 0.17	

As shown in Table 1, the approach presented in this article demonstrates outstanding performance on the MiniImageNet dataset. Leveraging ResNet-12 as the backbone network, the article compared the APFP method against several common few-shot learning approaches. Under the evaluation metric of 5-way 1-shot, the method proposed in this article achieves outstanding results, ranking second only to SetFeat. It outperformed Meta-DeepBDC and DMF by 0.54% and 0.22%, respectively. Additionally, under the 5-way 5-shot evaluation metric, the APFP method surpassed all other approaches, exhibiting a 0.86% improvement over Meta-DeepBDC. The exceptional performance of the APFP method can be attributed to the introduced FResNet and Adaptive Prototype, which enhance the model’s representational and generalization capabilities while deriving class prototypes that better align with sample features. Particularly noteworthy is the remarkable performance improvement of the APFP method under the 5-way 5-shot scenario, highlighting the significance of excellent prototype selection and embedding network construction for model performance.

As shown in Table 2, the proposed method in this article performs equally outstandingly on the CUB dataset. Comparative analyses were conducted between the APFP method and various popular methods. In the 5-way 1-shot scenario, the method proposed in this article falls slightly behind the BD-CSPN method, surpassing other methods, and achieves a 0.47% improvement over the baseline. In the 5-way 5-shot scenario, the APFP method proposed in this article outperforms all other methods, achieving a 0.62% improvement in accuracy compared to the baseline. Experiments on the CUB dataset demonstrate that the proposed method effectively enhances model accuracy in fine-grained dataset classification tasks, particularly excelling in the 5-shot scenario.

Tables 1 and 2 show that in the 5-shot scenario, the APFP method proposed in this article is significantly better than other prototype optimization methods, such as WPN, BD-CSPN, and DSFN. Firstly, the FResNet used in this article can more effectively extract sample features. Secondly, the Adpative Prototype method proposed in this article can dynamically calculate class prototypes based on query samples, effectively reducing the impact of biased samples on prototypes. In contrast, other methods use only a fixed class prototype and cannot adjust dynamically. Another advantage of the Adaptive Prototype method in this article is its lightweight nature, as it only needs to calculate the optimized class prototype based on the relationship between query samples and support samples. In contrast, many other methods require introducing new networks or modules. In summary, the APFP method in this article has more advantages.

Table 3 shows a detailed comparison between the APFP method proposed in this article and the baseline method DeepBDC regarding memory usage and runtime. Although the APFP method significantly outperforms the DeepBDC method in classification accuracy, it exhibits a disadvantage in memory usage and runtime. The memory usage and runtime of the APFP method are higher than those of the DeepBDC method, mainly due to its complex feature pyramid structure. While this structure helps extract richer features, it also increases the complexity of the model, leading to increased memory and time requirements. Additionally, the Adaptive Prototype method requires more complex calculations than the Mean Prototype when computing prototypes, further increasing the runtime. The DeepBDC method is suitable for applications with strict memory and time requirements. In contrast, the APFP method is undoubtedly a more appropriate choice for applications that demand higher classification accuracy despite its higher memory and time costs, as the improvement in accuracy may be worth it.

Table 3 Performance comparison on mini-ImageNet and CUB datasets.

Method	Mini-imagenet	CUB	
5-way 1-shot	5-way 5-shot	5-way 1-shot	5-way 5-shot	
Memory	Time	Memory	Time	Memory	Time	Memory	Time	
DeepBDC	47.5 M	95 s	47.5 M	118 s	39.6 M	86 s	39.6 M	101 s	
APFP	66.9 M	160 s	66.9 M	221 s	45.6 M	113 s	45.6 M	142 s	

By changing the parameter θ, a curve graph was generated to show the variation of classification accuracy with θ (Fig. 4). It was observed that the best classification accuracy was achieved when θ was set to 0.72. This indicates that at this parameter value, the calculation of class prototypes is more accurate, enabling a more effective differentiation of different categories in the dataset, thereby significantly improving overall classification performance. Additionally, the article conducted a visualization experiment on the FResNet feature extractor using the CUB dataset, as shown in Fig. 5. The distribution of data points was observed by mapping high-dimensional features to a two-dimensional space using a scatter plot. The experimental results show that the feature extractor can effectively separate features of different categories, demonstrating a clear clustering trend.

Figure 4 Classification accuracy on the CUB dataset with varying hyperparameter θ.

Accuracy of the 5-way 5-shot scenario on the CUB dataset as a function of the hyperparameter θ. The current hyperparameter θ value of 0.72 yields the best classification performance, with an accuracy of 94.44%.

Figure 5 Visualization of FResNet feature extraction.

Ablation study

In order to demonstrate the effectiveness of the two proposed methods in this article, the results of Meta-DeepBDC were reproduced in the experimental environment of this study using the source code and parameters provided by the authors of Meta-DeepBDC. Meta-DeepBDC was utilized as a baseline in further ablation studies to validate the efficacy of the proposed FResNet architecture and Adaptive Prototype method on the MiniImageNet dataset. Notably, disparities exist between the results reproduced in this article and those reported in the original study due to differences in experimental environments. The findings of the ablation experiments conducted in this study are presented in Table 4.

Table 4 Effectiveness of different modules on MiniImageNet.

It is noteworthy that the Adaptive Prototype method only applies in the 5-shot scenario, and is not discussed in the 1-shot scenario.

FResNet	Adaptive prototype	5-way 1-shot	5-way 5-shot	
×	×	66.82 ± 0.44	84.24 ± 0.28	
✓	×	68.09 ± 0.44	84.89 ± 0.27	
×	✓	—	84.39 ± 0.28	
✓	✓	68.09 ± 0.44	85.32 ± 0.28	

Effectiveness of FResNet. In this ablation study, the feature extractors of the original ResNet-12 architecture were contrasted with those derived from the ResNet-12-based FResNet. Upon substituting ResNet-12 with the FResNet method, the FResNet integrates features from different hierarchical levels during feature extraction, thereby retaining more detailed features and yielding feature maps with richer information content. As evident from Table 4, within the framework of Meta-DeepBDC, replacing ResNet with the FResNet proposed in this article resulted in a 1.17% enhancement in 5-way 1-shot classification by extracting additional information. Furthermore, under the 5-way 5-shot scenario utilizing Mean Prototype as the prototype calculation method, the efficacy of the FResNet proposed in this article was augmented by 0.65%. Substituting the Mean Prototype with the Adaptive Prototype proposed in this article as the prototype calculation method under the 5-way 5-shot scenario improved accuracy by 1.08%. This series of experimental findings distinctly underscores the substantial advantage of the FResNet structure proposed in this article in feature extraction, rendering it more effective than the conventional ResNet. Additionally, in the 5-shot scenario, the combination of FResNet with the Adaptive Prototype method yields even more outstanding performance. These experimental results comprehensively validate the effectiveness of the FResNet method proposed in this article and underscore its immense potential in the realm of few-shot learning.

Effectiveness of Adaptive Prototype. In this ablation study, the commonly used Mean Prototype method is contrasted with the Adaptive Prototype method proposed in this article. The Mean Prototype and Adaptive Prototype methods are utilized when the support set sample size exceeds one for computing class prototypes. Consequently, this experiment primarily focuses on comparing their efficacy under the 5-way 5-shot scenario, while the 5-way 1-shot scenario directly utilizes the respective sample as the class prototype and does not need further discussion. Within the Meta-DeepBDC framework, this article replaces the original Mean Prototype method with the Adaptive Prototype method. The Adaptive Prototype method computes more representative class prototypes based on query samples. As evident from Table 4, employing the Adaptive Prototype method proposed in this article resulted in a 0.15% improvement in 5-way 5-shot classification when the backbone network employs the ResNet-12 architecture and a 0.43% improvement when utilizing the FResNet structure. The experimental results in Table 3 demonstrate that the proposed Adaptive Prototype method can compute more representative class prototypes.

Conclusion

The APFP method, through the innovative FResNet feature extractor and Adaptive Prototype calculation method, effectively addresses the few-shot learning problem. FResNet combines the ResNet architecture with a feature pyramid structure, enhancing extracting local and global features from images to generate information-rich feature representations. The Adaptive Prototype method dynamically calculates class prototypes from the support set based on the query samples, overcoming the limitations of using simple sample means as prototypes and capturing class characteristics more accurately. These two innovations enable the APFP method to improve performance significantly across multiple few-shot learning benchmark datasets. Evaluations of the MiniImagenet and CUB datasets demonstrate that the proposed APFP method significantly outperforms previous approaches, achieving state-of-the-art performance on both datasets. The article provides an effective solution to the few-shot learning problem.

Additional Information and Declarations

Competing Interests

Author Contributions

Data Availability

The authors declare that they have no competing interests.

Linshan Shen conceived and designed the experiments, performed the experiments, analyzed the data, authored or reviewed drafts of the article, and approved the final draft.

Xiang Feng conceived and designed the experiments, performed the experiments, analyzed the data, performed the computation work, prepared figures and/or tables, and approved the final draft.

Li Xu conceived and designed the experiments, authored or reviewed drafts of the article, and approved the final draft.

Weiyue Ding performed the experiments, performed the computation work, prepared figures and/or tables, and approved the final draft.

The following information was supplied regarding data availability:

The CUB (Caltech-UCSD Birds 200) dataset was jointly created by Caltech and UC San Diego to promote research in bird identification and classification. The dataset contains images of 200 bird species and is suitable for image classification and fine-grained classification tasks. It is available at Wah, C., Branson, S., Welinder, P., Perona, P., & Belongie, S. (2022). CUB-200-2011 (1.0) [Data set]. CaltechDATA. https://doi.org/10.22002/D1.20098.

The MiniImageNet dataset was created by the Google DeepMind team in 2016 to support research on few-shot learning and meta-learning. This dataset is extracted from the large ImageNet dataset and contains diverse image samples, which is suitable for few-shot classification tasks. This dataset is currently the most commonly used dataset in the field of few-shot learning. Though there is no official repository available, it is available at figshare: feng, xiang (2024). miniImageNet.zip. figshare. Dataset. https://doi.org/10.6084/m9.figshare.26340226.v1.

The code is available at GitHub and Zenodo:

- https://github.com/fengxiang521/APFP

- Xiang, F. (2024). The code of APFP. Zenodo. https://doi.org/10.5281/zenodo.12799367.

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
