# Peer review of "Adaptive prototype few-shot image classification method based on feature pyramid"

_PeerJ Computer Science, doi:10.7717/peerj-cs.2322_

## Round 0.1 · original submission · Major Revisions

The authors must improve their article based on observations of reviewers.

Reviewer 1 ·

Basic reporting

The goal of few-shot learning is to enable machines to construct classifiers capable of recognizing unseen novel classes using only limited samples, akin to human ability.Metric learning-based approaches for few-shot classification represent a crucial methodology in addressing such challenges, with theirl performance primarily contingent upon the efficacy of feature extraction and prototype computation methodologies. The fundamental concept of metric learning involves utilizing feature extractors to extract target features, subsequently assessing similarity by comparing distances between prototypes. This paper introduces a novel feature extraction method, termed FResNet, which builds upon the ResNet architecture and leverages feature pyramid structures to retain finer details during computation.

Experimental design

This paper evaluates the accuracy of the proposed FResNet and Adaptive Prototype on the MinilmageNet and CUB datasets. Meta-DeepBDC is chosen as the baseline for comparison against recent popular few-shot classification methods, most of which fall under the category of metric-based methods. Two commonly used benchmarks, namely 5-way 5-shot and 5-way 1-shot, were selected for evaluation in this study. Experimental results are better.

Validity of the findings

To address this issue, an adaptive prototype method is proposed in this paper, which dynamically computes support set class prototypes based on the similarity between support set samples and query samples. Finally, the effectiveness of the proposed adaptive prototype few-shot image classification method based on feature pyramid (APFP) was validated through implementation on both the MinilmageNet dataset and the CUB dataset.

Additional comments

1. Add detailed indexes in abstract.,
2. Why use ResNet in this paper? please makes comparison with other related methods.
3. Some parameters are not given explanation.
4. Give more contents in experiments part.
5. Future works should be given to enhance the work.

Cite this review as

Reviewer 2 ·

Basic reporting

see the comments below

Experimental design

see the comments below

Validity of the findings

see the comments below

Additional comments

Few-Shot Image Classification is an emerging but promising research direction, which has far-reaching significance for some specific situations. In this paper, the authors also focused on this task and made a certain contribution, which is meaningful on the whole. But the reviewers still have some questions about the article that require the author to consider and elaborate, for instance:
1.Abbreviations should have clear and accurate correspondence with their full terms, such as Feature Pyramid and APFP in the abstract.
2.There are many methods to improve few-shot learning performance by refining prototypes or, more specifically, using multiple prototypes. What differentiates the framework proposed in this paper? A deeper analysis and comparison are highly necessary.
3.It is recommended to replace Figure 3 with real features rather than a toy example.
4.Even the best algorithms can fail in certain situations. Analyzing these cases helps readers gain a deeper understanding.
5.In addition, the application of few-shot learning techniques in other domains such as object detection, semantic segmentation, etc., should also be mentioned in the paper. The authors can refer to the following articles: spnet: siamese-prototype network for few-shot remote sensing image scene classification; holistic prototype activation for few-shot segmentation; base and meta: a new perspective on few-shot segmentation; few-shot segmentation via divide-and-conquer proxies.
6.It appears that the experiments in the paper primarily focus on classification accuracy, but algorithmic complexity (memory footprint, runtime, parameter count, etc.) is also a crucial evaluation metric that significantly impacts practical efficiency in real-world applications.
7.Correct the grammatical mistakes and polish them with native speakers if possible.

Cite this review as

---

## Round 0.2 · accepted · Accept

The paper can be accepted. It was well improved.

Reviewer 2 ·

Basic reporting

no comment

Experimental design

no comment

Validity of the findings

no comment

Additional comments

no comment

Cite this review as